# Role of Relebactam in the Antibiotic Resistance Acquisition in *Pseudomonas aeruginosa*: In Vitro Study

**DOI:** 10.3390/antibiotics12111619

**Published:** 2023-11-11

**Authors:** Maria Paz Ventero, Jose M. Haro-Moreno, Carmen Molina-Pardines, Antonia Sánchez-Bautista, Celia García-Rivera, Vicente Boix, Esperanza Merino, Mario López-Pérez, Juan Carlos Rodríguez

**Affiliations:** 1Microbiology Department, Dr. Balmis University General Hospital, Alicante Institute for Health and Biomedical Research (ISABIAL), 03010 Alicante, Spain; ventero_mar@isabial.es (M.P.V.); sanchez_antbau@gva.es (A.S.-B.); garcia_celriv@gva.es (C.G.-R.); rodriguez_juadia@gva.es (J.C.R.); 2Evolutionary Genomics Group, División de Microbiología, Universidad Miguel Hernández, Apartado 18, 03550 San Juan de Alicante, Spain; 3Institut de Biologie Structurale J.-P. Ebel, Université Grenoble Alpes, 38000 Grenoble, France; 4Infectious Diseases Unit, Dr. Balmis University General Hospital, Alicante Institute for Health and Biomedical Research (ISABIAL), 03010 Alicante, Spain

**Keywords:** relebactam, *Pseudomonas aeruginosa*, antibiotic resistance

## Abstract

Background: *Pseudomonas aeruginosa* shows resistance to several antibiotics and often develops such resistance during patient treatment. Objective: Develop an in vitro model, using clinical isolates of *P. aeruginosa*, to compare the ability of the imipenem and imipenem/relebactam to generate resistant mutants to imipenem and to other antibiotics. Perform a genotypic analysis to detect how the selective pressure changes their genomes. Methods: The antibiotics resistance was studied by microdilution assays and e-test, and the genotypic study was performed by NGS. Results: The isolates acquired resistance to imipenem in an average of 6 days, and to imipenem/relebactam in 12 days (*p* value = 0.004). After 30 days of exposure, 75% of the isolates reached a MIC > 64 mg/L for imipenem and 37.5% for imipenem/relebactam (*p* value = 0.077). The 37.5% and the 12.5% imipenem/relebactam mutants developed resistance to piperacillin/tazobactam and ceftazidime, respectively, while the 87.5% and 37.5% of the imipenem mutants showed resistance to these drugs (*p* value = 0.003, *p* value = 0.015). The main biological processes altered by the SNPs were the glycosylation pathway, transcriptional regulation, histidine kinase response, porins, and efflux pumps. Discussion: The addition of relebactam delays the generation of resistance to imipenem and limits the cross-resistance to other beta-lactams. The clinical relevance of this phenomenon, which has the limitation that it has been performed in vitro, should be evaluated by stewardship programs in clinical practice, as it could be useful in controlling multi-drug resistance in *P. aeruginosa*.

## 1. Introduction

Antibiotic resistance is one of the main challenges of Public Health worldwide. Particularly, *Pseudomonas aeruginosa* is a pathogen included within the six pathogens named with the acronym “ESKAPE” by Rice L.B in 2008 [1], to denote the main cause of life-threatening nosocomial diseases due to its virulence and high rates of resistance to the action of antibiotics. In addition, this pathogen often develops antibiotic susceptibility changes during treatment, especially in intensive care units and immunosuppressed patients [2,3]. The persistent and repeated use of antimicrobials in these critical units exerts selective pressure, hastening the emergence of resistant mutants [4]. Moreover, the transfer of resistance genes among these mutants can lead to the creation of “superbugs”—bacterial strains resistant to most clinically used antimicrobials [5]. Recognizing the urgency of the situation, healthcare providers and researchers have implemented stewardship initiatives. These initiatives encompass a wide range of measures, from the swift diagnosis through rapid microbiological techniques to tailored antibiotic therapies. By understanding these challenges and the ongoing efforts to address them, our study delves into the intricate dynamics of antibiotic resistance in *P. aeruginosa*, shedding light on crucial areas that demand exploration [6,7]. 

Also, the fight against antimicrobial resistance involves the development of new drugs or molecules that improve the susceptibility rates of current antimicrobials [8]. On this line, new coadjutants compounds have been developed, such as relebactam, which is a β-lactamase inhibitor with the ability to inhibit a broad spectrum of β-lactamases [9]. Apart from its antibiotic activity, it has been postulated it could be involved in the dynamics of the acquisition of imipenem resistance in *P. aeruginosa*, reporting that the addition of relebactam delays the appearance of the phenomenon in cultures of *P. aeruginosa* PAO1 (the most commonly used strain for research) [10]. 

To date, the efflux pump and the porins have been implicated in imipenem- and imipenem/relebactam-resistance in *P. aeruginosa* [11,12]. Previously, our research group performed a genomic study of 40 nosocomial strains of *P. aeruginosa* [13], and showed *P. aeruginosa* has a dynamic and highly plastic genome by which clinical strains of *P. aeruginosa* acquire resistance. These mechanisms allow bacteria to adapt quickly to unfavorable conditions, such as a selective pressure promoted by long antibiotic treatment [13]. 

To understand how prolonged exposure to imipenem or imipenem/relebactam influenced resistance generation against other antibiotics, and how the genome was modified under antibiotic selective pressure as an approximation, an in vitro model was performed. The model was based on the exposure of eight clinical isolates of *P. aeruginosa* to increasing concentrations of imipenem in the presence and absence of relebactam. Subsequently, the differences between the two conditions were analyzed by phenotypic characterization against different antibiotic families and genotypic characterization, using NGS (Next-Generation Sequencing) based on short reads (Illumina), and long reads (PacBio). Both techniques were used because Illumina, although the most commonly used, requires the genome to be broken into small fragments, which increases bias during bioinformatic analysis. However, PacBio allows sequencing without fragmentation, so the bias in the bioinformatic analysis was reduced.

## 2. Results

### 2.1. Resistance Acquisition to Imipenem and Imipenem/Relebactam

Initially, all strains showed a minimum inhibitory concentration (MIC) ≤ 1 mg/L for imipenem, and a MIC ≤ 0.25 mg/L for the combination of imipenem/relebactam. After repeated exposure by subculturing, isolates acquired resistance (MIC > 4 mg/L) to imipenem in an average time of 6 days, and to the imipenem/relebactam (MIC > 2 mg/L) in an average time of 12 days according to the threshold established by the European Committee on Antimicrobial Susceptibility Testing (EUCAST) [14]. This difference in resistance acquisition time was statistically significant (*p* value = 0.004). However, the time to acquire resistance at higher concentrations (MIC ≥ 32 mg/L) was not significant (*p* value = 0.504) between the two treatments. On day 18, the imipenem-treated strains achieved a mean MIC above 32 mg/L, and on day 20, the imipenem/relebactam-treated isolates achieved a mean MIC above 32 mg/L. Finally, after 30 days of exposure to the antibiotic, 75% of the clinical isolates (6/8) reached a MIC > 64 mg/L for imipenem and only 37.5% (3/8) for imipenem/relebactam (*p* value = 0.077) (Figure 1). 

### 2.2. Resistance Acquisition to the Other Antibiotics Tested by the Mutants

The eight initial isolates were susceptible to all the antibiotics tested (beta-lactams, aminoglycosides, and quinolones) according to the MIC established by the EUCAST. 

Among the five beta-lactams tested, 100% (8/8) of the imipenem-resistant mutants acquired resistant to at least one of them, 50% (4/8) acquired resistance to three of the beta-lactams tested, and 25% (2/8) to all of them. In contrast, for the imipenem/relebactam-resistant mutants, the resistance rates decreased. Only 50% were resistant for one, 12.5% for three of them, and none of them for the five tested. When comparing the acquisition of resistance between mutants based on the mean MIC obtained after the in vitro experiment, the imipenem/relebactam-resistant mutants achieved a lower mean MIC than the imipenem-resistant mutants for piperacillin/tazobactam (17.75 mg/L ± 3.73; 198 mg/L ± 87.50) and ceftazidime (4.81 mg/L ± 1.16; 77.87 mg/L ± 39.06) (*p* value = 0.003 and *p* value = 0.015, respectively). For the other antibiotics tested, there was no significant difference between the two groups of mutants (Table 1, Table 2, and Appendix A). 

### 2.3. Genotypic Characterization of the Initial Isolates and Resistant Mutants to Imipenem and Imipenem/Relebactam

The initial isolates (8247 and 2718) were sequenced by short and long read technologies (Illumina NextSeq and PacBio Sequel II, respectively). The genomes were assembled into a single sequence (without plasmids) of ca. 6.5 Mb and a GC of 65%, as well as a coding density of 90% (Table 3). Average nucleotide identity (ANI) between the two strains was 98.84% and according to the multilocus sequence typing (MLST) analysis, 8247 and 2718 belonged to the ST253 and ST238 type sequences, respectively, based on the pubMLST database [15]. 

Once the genomes were assembled, we analyzed the intrinsic genomic capacity of the initial strains for antibiotic resistance by identifying resistance genes with the Comprehensive Antibiotic Resistance Database (CARD) [16]. After manual curation, we found two OXA-50 (oxacillinase, β-lactamase) variants (OXA-846 and OXA-488) and PDC (Purified β-lactamases) variants (PDC-127 and PDC-34) in both strains; these genes are intrinsic resistance genes in *P. aeruginosa* (Table 3). 

Next, we aligned the Illumina raw reads from the sequencing of the resistance strains (MIC > 64 mg/L) over the assembled genome of the sensitive strains to detect the genomic variants generated for both antibiotics (imipenem and imipenem/relebactam). Imipenem-resistant mutants presented two and nine single nucleotide polymorphisms (SNPs, changes of one nucleotide), compared to their respective susceptible strains (Table 4 and Appendix A). We evaluated the effect of the detected SNPs on coding sequences, that is, missense (the SNPs produce a different amino acid being incorporated into the structure of the protein), frameshift (reading frame changes, resulting in a protein completely different from the original), and stop codon mutations resulting in a smaller, usually non-functional protein, as well as in non-coding regions, which may alter the transcription of the closest gene. While strain 8247 showed only significant variations on the gene ydphP_1, in the other strain (2718), seven were the genes in which non-synonymous variations (missense change) were produced (*phoQ*, *gyrB*, *wecA*, *sasA_7*, *nuoM*, *oprD_7*, *pilA*). On the other hand, imipenem/relebactam -resistance mutant showed five and eighteen total SNPs when compared with the sensible strain. In this case, genes with significant variants were *rne*, *sasA_14*, *qseC_2* in 8247 strain, and *tyrR_1*, *rstB*, *barA_1*, *zrarR_4*, *pglF*, *rne* and *rnr*, in 2718 strain. Strain 8247 showed convergent variation in gene *tagO* when treated with both treatments, so this SNP could be shared in the imipenem and imipenem/relebactam resistance acquisition mechanism. 

Despite not obtaining mutations in the same genes, analysis of biological processes revealed the involvement of several pathways in common with both treatments. For instance, glycosylation pathways (*pglF*, *tagO* and *wecA*), two-component system sensor histidine kinase (*sasA*, *qseC*, *barA*), and porins (*rstB*, *oprD_7*) were altered in the mutans generated with both treatments. 

Regarding the pathways affected by each of the treatments individually, the imipenem-resistant mutants showed alterations in pathways related to resistance to other antibiotics such as chloramphenicol (*ydphP*), aminoglycosides (*phoQ*), and fluoroquinolones (*gyrB*), as well as genes related to ATP synthesis (*nuoM*) and biofilm formation (*pilA*). In contrast, the several transcriptional regulation genes (*rne*, *tyrR*, *zrarR*, *rnr*) were only affected in the imipenem/relebactam-resistant mutants (Table 4).

In summary, the findings indicate the selective pressure exerted by prolonged exposure to imipenem involves changes at the genotypic level in several biological pathways, and when relebactam is added, the changes also affect the transcriptional mechanism.

## 3. Discussion

The resistance to carbapenems and to other antibiotics is increasing during the last years, especially in *P. aeruginosa*. Likewise, the mortality associated with infections caused by this pathogen has also become a growing threat [17,18]. To improve the carbapenems’ activity, several compounds have been developed which are administered together with the antibiotic, for example, relebactam, a beta-lactamase inhibitor [9,19,20,21]. This work shows the relebactam addition delays the generation of antibiotic resistance to imipenem in *P. aeruginosa*. At the same time, the addition of relebactam limits the emergence of cross-resistance to other drugs, a key phenomenon in controlling antibiotic resistance in certain hospital areas, where a situation similar to the one generated in vitro in this article exists, due to the massive use of carbapenems. 

Regarding the genomic analysis, Gomis-Font MA et al. [10] found mutations in *OprD*, in *ampC* and *MexAB-OprM* regulators in strains of *P, aeruginosa*. In line with this, our study identified mutations in efflux pump genes, such as *MexAB-OprM* regulators. These mutations are known to enhance the bacteria’s resistance mechanisms by limiting antibiotic efflux, a crucial aspect of resistance development. Mainly, the mutated genes shared by the two types of mutants are involved in efflux pumps, glycosylation pathway, and histidine kinase response. The relation of the efflux pumps with antibiotic resistance is widely studied [22,23], and the role of the porin OprD and the *MexXY/OprM*, which are involved in the PaRS two-component regulation system from *P. aeruginosa* [24,25,26], homologous to *RstA* and *RstB* in *P. fluorescens* [27], is clearly related to the multidrug-resistant acquisition [28,29]. Besides, the efflux pumps are usually regulated by the denominated two-component regulatory system, as mentioned above, this system consists of an environmental sensor (histidine kinase), and an intracellular control factor response regulator [30], and is also related to antibiotic resistance [31]. In a previous study, we postulated that the processes leading to imipenem/relebactam resistance were multifactorial or that regulation occurred at the transcriptional level, exerting its effect on gene expression [13]. This evidence is in line with the results obtained because the SNPs were located in genes involved in both resistance mechanism (*sasA*, *qseC*, *phoQ*, *barA*, *rstB*, *oprD_7*). On the other hand, the glycosylation pathway was also affected by the SNPs detected (*pglF*, *tagO* and *wecA*). This pathway is closely related to the lipopolysaccharide (LPS), the major component of the outer membrane of Gram-negative bacteria [32], and it has been shown that it could be related to antibiotic resistance [33,34], and even the modification of the bacteria’s glycosylation has been proposed as a novel antibacterial strategies [35]. In addition, this work shows imipenem-resistant mutants have altered pathways directly related to antibiotic resistance (Chloramphenicol resistance, resistance to Polymyxin B and cationic antimicrobial peptides, resistance to fluoroquinolones and biofilm formation). However, only the imipenem/relebactam-resistant mutants modified genes related to transcriptional regulation (*rne*, *tyrR*, *zrarR*, *rnr*), this mechanism is also related to changes in antimicrobial resistance [36,37]. 

Genotypic characterization was not conducted on all strains due to resource constrainsIn addition, the gDNA was isolated from one single colony for each strain, so it is important to acknowledge this omission might introduce biases in our understanding of the genetic basis of resistance development. Future studies should consider addressing this limitation by increasing the number of strains sequenced and isolating the gDNA from different colonies of each strain, to provide a more comprehensive analysis. This genomic analysis was proposed as a preliminary sub-study on how prolonged and repeated exposure to a drug generates changes at the genomic level and to explore at the genotypic level the differences in resistance acquisition found at the phenotypic level. However, the SNPs found in the transcriptional regulation pathway highlight a limitation for this study because the antibiotic resistance acquisition could be mediated by changes in the gene expression profile, and using WGS (Whole Genome Sequencing), these changes cannot be detected. For this reason it would be interesting to follow this experiment with an RNA-seq study, as in the study performed by Cianciulli Sesso, where they hypothesized the resistance to Colistin and Tobramycin could be related to the expression of several genes [38]. 

Based on the data obtained, the short-duration treatments with the imipenem/relebactam, even if the microorganism is susceptible to carbapenems, could prevent the appearance of resistance during treatment. However, while proposing short-duration treatments with imipenem/relebactam, it is crucial to consider the practical challenges of administering such regimens in clinical settings. Factors such as patient compliance, dosing schedules, and potential side effects should be thoroughly evaluated to determine the feasibility and effectiveness of this approach. Today, the multidrug resistance in *P. aeruginosa* is a first-level public health problem, both sanitary and economic [39,40,41], and this challenge must be addressed through antimicrobial stewardship groups in order to achieve the appropriate use of these drugs and prevent resistance to them [42,43,44].

## 4. Materials and Methods

### 4.1. Bacterial Strains

Eight clinical isolates of *P. aeruginosa* from the collection of the Microbiology Service of the Hospital General Universitario Dr. Balmis de Alicante (Spain) were studied, without any epidemiological association between them. All the strains were susceptible to imipenem and imipenem/relebactam (EUCAST 2023 criteria), isolated from respiratory, urine, or wound exudate samples during 2020. 

### 4.2. In Vitro Generation of P. aeruginosa Resistant Mutants

The initial inoculum for the continuous antibiotic exposure experiment was 200 µL of a 0.5 McFarland suspension of each *P. aeruginosa* isolate. These 8 inocula were seeded in 96-well plates contained imipenem or imipenem/relebactam. Every day for 30 days, 100 µL from the last well with observable growth, after incubation, was used as inoculum for the next seeding on a new plate with the same concentration of antibiotic. For all passages, the antibiotic concentrations for imipenem were 0.125, 0.25, 0.5, 1, 2, 4, 8, 16, 32, 64, and 128 mg/L. For the model with relebactam, the same concentrations of imipenem were used, and relebactam was added at a fixed concentration of 4 mg/L according to EUCAST guidelines. The incubation for each passage was at 37 °C for 24 h. At the end of the experiment, eight mutants resistant to imipenem, and eight mutants resistant to imipenem/relebactam were obtained; to ensure the stability of the mutations generated, the strains were re-seeded five consecutive times on antibiotic-free medium.

### 4.3. Antibiotic Susceptibility

Susceptibility of the initial strains and their resistant mutants generated was assessed using Epsilometer test (E-test) (Biomérieux, Marcy-l’Étoile, Francia). The antibiotic studied were beta-lactams: piperacillin/tazobactam (Pfizer, New York, NY, USA, EEUU), ceftazidime (Pfizer, New York, EEUU), ceftolozane/tazobactam (MSD, Rahway, NJ, USA, EEUU), ceftazidime/avibactam (Pfizer, New York, EEUU), and aztreonam (Pfizer, New York, EEUU), aminoglycosides: amikacin (Normon, Madrid, Spain), and tobramycin (Pfizer, New York, EEUU), and quinolones: ciprofloxacin (Pfizer, New York, EEUU). The results obtained from E-test were interpreted according to the EUCAST 2023 protocol (Clinical breakpoints-bacteria (v 13.0) [14]. 

### 4.4. Genotypic Characterization and Bioinformatic Analyses

From the initial eight strains, two of them were randomly selected to carry out whole-genome sequencing (8247 and 2718). Multi Locus Sequence Typing (MLST) was performed using ARIBA (Antimicrobial Resistance Identification By Assembly) [45], with the pubmlstget option to get the sequence type classification of *P. aeruginosa* from the PubMLST database [15]. The strains sequenced were (i) the initial isolates (imipenem and imipenem/relebactam susceptible), (ii) the imipenem-resistant mutants generated (MIC > 4 mg/L), and (iii) the imipenem/relebactam-resistant mutants (MIC > 2 mg/L). Strains were sequenced by Illumina NextSeq 500/550 High Output Kit v2.5–300 Cycles, using the kit DNA Prep distributed by Illumina (ISABIAL, Hospital General Universitario Dr. Balmis de Alicante, Spain). In addition, the genomes of the initial strains were also sequenced by PacBio Sequel II (FISABIO, Valencia, Spain) to obtain closed genomes in a single sequence to be used as references for downstream analyses (see below). Illumina raw reads were trimmed with Trimmomatic v0.39 [46] to remove bad quality and remnant adapter sequences. On the other hand, Highly Accurate Single-Molecule Consensus Reads (CCS reads) were generated from the PacBio raw data, using the CCS v6.2 program of the SMRT-link package. Reference genomes (i.e., susceptible strains) were assembled using Flye v2.9 [47] using the following options: –pacbio-hifi –genome-size 6.6 m. Then, assembled contigs were corrected using Pilon [48] using the Illumina trimmed reads. Prodigal v2.6.3 [49] was used to predict genes from the assembled contigs. Predicted protein-encoded genes were taxonomically and functionally annotated against the NCBI NR database using DIAMOND v0.9.15 [50] and against COG [51] and TIGRFAM [52] using HMMscan v3.3 [53]. Lastly, determination of single-nucleotide polymorphisms (SNPs) and variant calling were determined by Snippy (https://github.com/tseemann/snippy, accessed on 30 August 2023) by aligning the intermediate and resistant Illumina sequencing data to the initial reference genome with the following parameters: –basequal 20–minfrac 0.8–mincov 10. 

### 4.5. Statistical Analysis

Categorical variables were expressed as total number of strains or percentage. Mean or median was used for continuous variables. Statistical differences between groups were determined by Chi-square test or Fisher’s test for non-continuous variables, and Student’s *t*-test or Mann-Withney U-test for continuous variables. The tests used are two-tailed (split the significant level and applied in both direction), suitable for determining whether there is any difference between the groups and a *p*-value < 0.05 was required to consider significant statistical differences. Analyses were performed with SPSS Statistics software (IBM, version 23.0).

## Figures and Tables

**Figure 1 antibiotics-12-01619-f001:**
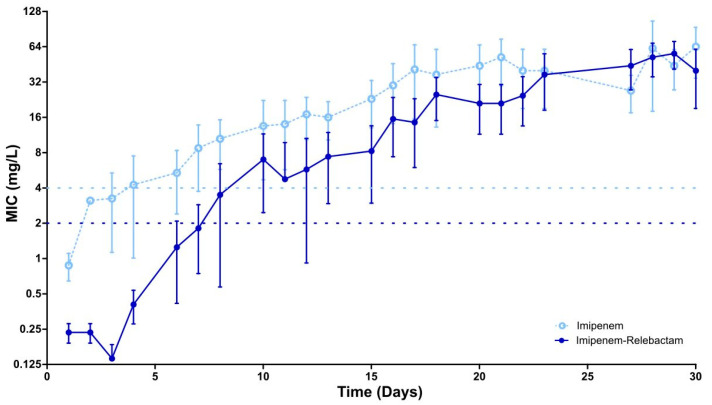
Mean MIC (mg/L) of the imipenem and imipenem/relebactam-resistant mutants along the continuous exposure. The mean MIC and the standard deviation bars were calculated from the eight strains, using them as biological replicates.

**Table 1 antibiotics-12-01619-t001:** Mean MIC (mg/L) and standard error for the initial isolates, and the mutants generated by continuous exposure to imipenem and to imipenem/relebactam, with the *p* value obtained for the mean MIC comparison between imipenem and imipenem/relebactam mutants.

		Initial Isolates		Imipenen Mutants		Imipenem/Relebactam Mutants		
		MIC (mg/L)	% R *	MIC (mg/L)	% R *	MIC (mg/L)	% R *	*p* Value
**Beta-lactams**	Piperacillin/ Tazobactam	4.75 ± 0.72	0	198 ± 38.00	87.5	17.75 ± 3.73	37.5	0.003
	Ceftazidime	0.62 ± 0.11	0	77.87 ± 39.06	37.5	4.81 ± 1.16	12.5	0.015
	Ceftolozane/ Tazobactam	0.39 ± 0.05	0	2.87 ± 0.93	25.0	1.81 ± 0.91	0	0.823
	Ceftazidime/ Avibactam	0.87 ± 0.09	0	38.75 ± 31.09	37.5	6.25 ± 1.61	12.5	0.442
	Aztreonam	2.56 ± 0.29	0	85 ± 37.73	62.5	18.25 ± 3.88	50	0.105
**Aminoycosides**	Amikacin	3.31 ± 0.31	0	13.12 ± 3.65	25.0	24.75 ± 10.71	50	0.442
	Tobramycin	0.87 ± 0.07	0	1.31 ± 0.16	0	1.84 ± 0.46	37.5	0.999
**Quinolones**	Ciprofloxacin	0.083 ± 0.01	0	0.10 ± 0.01	0	0.61 ± 0.19	50	0.279

* % R: Percentage of resistant strains to the antibiotics tested. MIC for considered resistant according to EUCAST 2023: Piperacillin/tazobactam (>16 mg/L), Ceftazidime (>8 mg/L), Ceftolozane/tazobactam (>4 mg/L), Ceftazidime/avibactam (>8 mg/L), Aztreonam (>16 mg/L), Amikacin (>16 mg/L), Tobramycin (>2 mg/L), Ciprofloxacin (>0.5 mg/L).

**Table 2 antibiotics-12-01619-t002:** Imipenem and imipenem/relebactam-resistant mutants which acquired resistance to other antibiotics *.

		Imipenem-Resistant Mutants	Imipenem/Relebactam-Resistant Mutants
**Beta-lactams**	Piperacillin/	3835, 8247, 6760, 9137, 6630, 2718, 3664	6630, 2718, 3664
	Tazobactam		
	Ceftazidime	3835, 6760, 9137, 6630, 2718, 3664	6630
	Ceftolozane/	6760, 9137	
	Tazobactam		
	Ceftazidime/	6760, 9137, 2718	6630
	Avibactam		
	Aztreonam	3835, 6760, 9137, 6591, 2718	8247, 6630, 2718, 3664
**Aminoycosides**	Amikacin	6760, 2718	6760, 6591, 6630, 2718
	Tobramycin		6760, 6591, 2718
**Quinolones**	Ciprofloxacin		3835, 9137, 6630, 3664

***** According to EUCAST 2023.

**Table 3 antibiotics-12-01619-t003:** Genomic features of the strains sequenced.

	Contigs	Genome Size (bp)	GC Content (%)	Proteins	Coding Density (%)	Intrinsic Antibiotic Resistance
**Strain 2718**	1	6,445,503	66.4	5915	90	OXA-846/PDC-127
**Strain 8247**	1	6,942,622	65.9	6457	89	OXA-488/PDC-34

**Table 4 antibiotics-12-01619-t004:** Variants with effect identified in the resistant mutants when compared with their initial isolates.

	Gene	I-R * Mutants	I/R-R * Mutants	Type	Effect	Protein	Pathway
**Strain 8247 (ST253)**	* tagO *	Yes	Yes	Del	frameshift	putative undecaprenyl-phosphate N-acetylglucosaminyl 1-phosphate transferase	Glycosylation mechanism
	*rne*	*No*	*Yes*	*Del*	*disruptive*	*Ribonuclease E*	*Transcriptional regulation*
	** *ydhP_1* ** ** *(cmxA)* **	**Yes**	**No**	**SNP**	**missense**	**Inner membrane transport protein YdhP**	**Chloramphenicol resistance**
	* sasA_14 *	No	Yes	SNP	stop	Adaptive-response sensory-kinase SasA	histidine kinase response system
	* qseC *	No	Yes	SNP	missense	Sensor protein QseC	histidine kinase response system
**Strain 2718 (ST238)**	*tyrR_1*	*No*	*Yes*	*SNP*	*missense*	*Transcriptional regulatory protein TyrR*	*Transcriptional regulation*
	* rstB *	No	Yes	SNP	missense	Sensor protein RstB	Efflux pump
	* barA_1 *	No	Yes	SNP	missense	Signal transduction histidine-protein kinase BarA	Histidine kinase response system
	** *phoQ* **	**Yes**	**No**	**SNP**	**missense**	**Two-component sensor PhoQ**	**Polymyxin B and cationic antimicrobial peptides resistance**
	*zraR_4*	*No*	*Yes*	*DEL*	*frameshift*	*Transcriptional regulatory protein ZraR*	*Transcriptional regulation*
	** *gyrB* **	**Yes**	**No**	**SNP**	**missense**	**DNA gyrase subunit B**	**Fluoroquinolones resistance**
	* wecA *	Yes	No	SNP	frameshift	Undecaprenyl-phosphate alpha-N-acetylglucosaminyl 1-phosphate transferase	Glycosylation mechanism
	* sasA_7 *	Yes	No	SNP	missense	Adaptive-response sensory-kinase SasA	histidine kinase response system
	** *nuoM* **	**Yes**	**No**	**DEL**	**disruptive**	**NADH-quinone oxidoreductase subunit M**	**ATP synthesis**
	* oprD_7 *	Yes	No	SNP	stop_gained	Porin D	Efflux pump
	** *pilA* **	**Yes**	**No**	**DEL**	**frameshift**	**Fimbrial protein**	**Biofilm formation**
	* pglF *	No	Yes	SNP	missense	UDP-N-acetyl-alpha-D-glucosamine C6 dehydratase	Glycosylation mechanism
	*rne*	*No*	*Yes*	*SNP*	*missense*	*Ribonuclease E*	*Transcriptional regulation*
	*rnr*	*No*	*Yes*	*SNP*	*stop_gained*	*Ribonuclease R*	*Transcriptional regulation*

* I-R mutants: imipenem-resistant mutant. * IR_R mutants: imipenem/relebactam-resistant mutant. Underlined sentences: pathways affected in both types of mutants. **Bold sentences: pathways affected only in imipenem-resistant mutants**. No special format sentences: pathways affected only in imipenem/relebactam-resistant mutants. The proteins have been assigned to the different pathways using UniProtKB Database and KEGG PATHWAY Database.

## Data Availability

The genomes obtained by PacBio and the raw data from Illumina have been deposited under BioProject PRJNA873916 in the NCBI Database.

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
