# Peer review of "Role of Relebactam in the Antibiotic Resistance Acquisition in Pseudomonas aeruginosa: In Vitro Study"

_antibiotics, 2023, doi:10.3390/antibiotics12111619_

Round 1

Reviewer 1 Report

Comments and Suggestions for Authors

Pseudomonas aeruginosa is one of the ESKAPE pathogens that significantly contribute to life-threatening nosocomial infections in healthcare settings and demonstrate multidrug resistance. The increased and often improper use of antibiotics in clinical settings is believed to contribute to the rising resistance of P. aeruginosa to multiple antibiotics. In this study, the authors investigated the impact of exposing P. aeruginosa to imipenem, with or without relebactam, on the development of resistance to these antibiotics as well as to others. The topic of the study is important in the field of antibiotic resistance. However, I have the following comments that should be addressed.

1. In the genotypic analysis, the authors utilized two imipenem-susceptible strains to develop strains resistant to imipenem and imipenem/relebactam. Whole genome sequencing was conducted to identify mutations conferring resistance to imipenem and other antibiotics, and to assess how the addition of relebactam affected these results. However, it is not clear whether gDNAs for sequencing were obtained from single colonies or a pool of bacterial cells carrying mutations in different locations after the acquisition of resistance strains. Moreover, a single experiment might not provide sufficient grounds for conclusive results. it is crucial to conduct experiments multiple times to ensure the reproducibility of the results.

2. In Figure 1, the legend should provide additional information. Specifically, it should clarify the nature of the error bars and confirm whether the Mean MIC was calculated from data obtained from 8 strains. Additionally, the number of replicates per strain used for this experiment should be mentioned.

3. In Table 1, including a column indicating the class of antibiotics, as demonstrated in Table S2, would enhance the reader's understanding of the results and facilitate following the data.

4. On lines 149 and 153, there is a query regarding why tagO was not included. The table indicates 'Yes' in both conditions; however, an explanation for this is needed.

5. I'm unsure whether lines 73 to 77 and lines 79 to 83 present the same information. Could the authors confirm if these sections contain similar data, and if so, provide an explanation for the differing days?

6. In Table 2, more information should be included in the table footnote. Particularly, the authors should elaborate on the pathway mentioned and provide additional details on how the pathway was assessed.

Author Response

Manuscript ID: antibiotics-2658806

Title: "Role of Relebactam in the antibiotic resistance acquisition in P. aeruginosa: in vitro study”

Reviewer #1:

Pseudomonas aeruginosa is one of the ESKAPE pathogens that significantly contribute to life-threatening nosocomial infections in healthcare settings and demonstrate multidrug resistance. The increased and often improper use of antibiotics in clinical settings is believed to contribute to the rising resistance of P. aeruginosa to multiple antibiotics. In this study, the authors investigated the impact of exposing P. aeruginosa to imipenem, with or without relebactam, on the development of resistance to these antibiotics as well as to others. The topic of the study is important in the field of antibiotic resistance. However, I have the following comments that should be addressed.

  1. In the genotypic analysis, the authors utilized two imipenem-susceptible strains to develop strains resistant to imipenem and imipenem/relebactam. Whole genome sequencing was conducted to identify mutations conferring resistance to imipenem and other antibiotics, and to assess how the addition of relebactam affected these results. However, it is not clear whether gDNAs for sequencing were obtained from single colonies or a pool of bacterial cells carrying mutations in different locations after the acquisition of resistance strains. Moreover, a single experiment might not provide sufficient grounds for conclusive results. it is crucial to conduct experiments multiple times to ensure the reproducibility of the results.

Response:

Before isolating the gDNAs, the strains were re-seeded five consecutive times on antibiotic-free medium to ensure that the mutations were stable, from the last culture a single colony was re-seeded to obtain sufficient biomass to isolate the gDNAs, and MALDI-TOFF identification and antibiotic susceptibility were also checked again.

  1. In Figure 1, the legend should provide additional information. Specifically, it should clarify the nature of the error bars and confirm whether the Mean MIC was calculated from data obtained from 8 strains. Additionally, the number of replicates per strain used for this experiment should be mentioned.

Response:

In this study, the 8 strains were used as biological replicates, as the strains had the same initial characteristics, therefore no technical replicates were included. We added this information in the Figure’s 1 legend.

The mean MIC and the error bars were calculated from the eight strains, using them as a biological replicates.”

  1. In Table 1, including a column indicating the class of antibiotics, as demonstrated in Table S2, would enhance the reader's understanding of the results and facilitate following the data.

Response:

This Column has been included.

  1. On lines 149 and 153, there is a query regarding why tagO was not included. The table indicates 'Yes' in both conditions; however, an explanation for this is needed. 

Response

An explanation has been added:

“Strain 8247 showed the convergent variation in gene tagO when treated with both treatments, so this SNP’s could be shared in the imipenem and imipenem/relebactam resistance acquisition mechanism.”

  1. I'm unsure whether lines 73 to 77 and lines 79 to 83 present the same information. Could the authors confirm if these sections contain similar data, and if so, provide an explanation for the differing days?

Response

This a writing error, really in this sentence the authors wanted to show the average of needed days to acquire a high-level resistance (MIC≥ 32 mg/L). The paragraph has been modified:

“On day 18, the imipenem-treated strains achieved a mean above  32 mg/L, and on day 20, the Imipenem/relebactam-treated isolates achieved a mean above 32 mg/L”

  1. In Table 2, more information should be included in the table footnote. Particularly, the authors should elaborate on the pathway mentioned and provide additional details on how the pathway was assessed.

Response

The footnote has been modified including information about the pathways assignation.

“The proteins have been assigned to the different pathways using UniProtKB Database and KEGG PATHWAY Database”

Reviewer 2 Report

Comments and Suggestions for Authors

good research

Title :
Role of Relebactam in the antibiotic resistance acquisition in Pseudomonas
aeruginosa: in vitro study
Abstract
Added the used methods
Introduction
What mean this abbreviation? using NGS based on
Discussion
homolog to RstA and RstB in P. fluorescens [35]
What mean this abbreviation? and using WGS these changes cannot detected.
4.3. Antibiotic susceptibility
Indicate origin of antibiotics, company.

Author Response

Reviewer #2:

Good research

Title:
Role of Relebactam in the antibiotic resistance acquisition in Pseudomonas aeruginosa: in vitro study

Response:

The title has been modified according the reviewer suggests.

Abstract
Added the used methods

Response:

The abstract has been completed:

“Methods: The antibiotics resistance was studied by microdilution assays and e-test, the genotypic study was performed by NGS.”

Introduction
What mean this abbreviation? using NGS based on

Response:

The abbreviation has been specified:

“using NGS (Next-Generation Sequencing) based on..”

Discussion
homolog to RstA and RstB in P. fluorescens [35]

Response:

The format of P. fluorescens has been change, now is in italic. 

What mean this abbreviation? and using WGS these changes cannot detected.

Response

The abbreviation has been specified:

“using WGS (Whole Genome Sequencing) these changes cannot….”

4.3. Antibiotic susceptibility. Indicate origin of antibiotics, company.

Response:

The companies has been included

“(piperacillin/tazobactam (Pfizer), ceftazidime (Pfizer), ceftolozane/tazobactam (Merk), ceftazidime/avibactam (Pfizer) and aztreonam (Pfizer)), aminoglycosides (amikacin (Normon) and tobramycin (Pfizer)) and quinolones (ciprofloxacin (Pfizer ).”

Reviewer 3 Report

Comments and Suggestions for Authors

1.    Abstract

The abstract presents a focused study on antibiotic resistance in Pseudomonas aeruginosa, a significant clinical concern. The research aimed to develop an in vitro model using clinical isolates of P. aeruginosa to compare the ability of imipenem and imipenem/relebactam in generating resistant mutants, along with a genotypic analysis to understand the genetic changes under selective pressure. Results indicate that isolates acquired resistance to Imipenem within an average of 6 days, whereas resistance to imipenem/relebactam occurred in 12 days (p-value=0.004). After 30 days, 75% of isolates had a minimum inhibitory concentration (MIC) > 64 mg/L for imipenem, and 37.5% for imipenem/relebactam (p-value=0.077). Notably, the addition of relebactam delayed resistance to imipenem and limited cross-resistance to other beta-lactams. Specific genetic alterations, particularly in glycosylation pathways, transcriptional regulation, histidine kinase response, porins, and efflux pumps, were identified. The study highlights the potential clinical significance of these findings, suggesting that stewardship programs should evaluate the implications for controlling multi-drug resistance in P. aeruginosa. Consideration of study limitations, such as the in vitro nature of the experiment and its applicability to clinical contexts, would further strengthen the abstract.

2.    Introduction

The introduction provides a concise overview of the antibiotic resistance issue, focusing on the critical case of Pseudomonas aeruginosa. It effectively emphasizes the urgent need for innovative therapeutic strategies due to the pathogen's adaptability and antibiotic resistance. The link between extended antibiotic usage and the emergence of resistant mutants is clearly articulated, highlighting the study's significance. To enhance clarity, briefly mentioning successful examples of stewardship initiatives and explicitly outlining the unique contribution of this research would add depth. Creating a smoother transition between topics could further improve the introduction's flow, making it more accessible and engaging for a wider readership.

Eg.

       Original

The persistent and repeated use of antimicrobials in these critical units exerts selective pressure, hastening the emergence of resistant mutants [3]. Moreover, the transfer of resistance genes among these mutants can lead to the creation of "superbugs" – bacterial strains resistant to most clinically used antimicrobials [4]. In response to this escalating crisis, stewardship initiatives are implementing diverse measures, ranging from rapid microbiological diagnostics to tailored antibiotic therapies, aiming to curb the rise of antibiotic resistance [5,6].”

       Revised with a Smoother Transition

“The persistent and repeated use of antimicrobials in these critical units exerts selective pressure, hastening the emergence of resistant mutants [3]. Moreover, the transfer of resistance genes among these mutants can lead to the creation of "superbugs" – bacterial strains resistant to most clinically used antimicrobials [4]. Recognizing the urgency of the situation, healthcare providers and researchers have implemented stewardship initiatives. These initiatives encompass a wide range of measures, from the swift diagnosis through rapid microbiological techniques to tailored antibiotic therapies. By understanding these challenges and the ongoing efforts to address them, our study delves into the intricate dynamics of antibiotic resistance in P. aeruginosa, shedding light on crucial areas that demand exploration.”

 As it connects the discussion of superbugs and stewardship initiatives, leading seamlessly into the purpose and relevance of the current study. This approach helps readers follow the logical progression of the introduction more easily.

3.Results

The results section of the research paper is well-structured, providing a clear and detailed account of the experiments conducted. The data are presented logically, and the use of tables and figures enhances the overall clarity of the results. The authors have appropriately utilized statistical analyses, supporting the presented data with relevant p-values to indicate the significance of the findings. However, it is important to explicitly mention the methods employed for the statistical analysis to ensure transparency and reproducibility.

       Genotypic Characterization and Biological Significance

The genotypic characterization section is detailed and informative, shedding light on the genetic basis of resistance. The description of the pathways affected by mutations is clear. However, considering the complexity of genetic terminology, providing brief explanations or references for readers less familiar with these terms could enhance the accessibility of the content. Moreover, it would be valuable to include a brief concluding paragraph summarizing the key findings of the genotypic characterization, helping readers grasp the implications of the genetic mutations in the context of antibiotic resistance.

In summary, the results section provides a detailed account of the resistance acquisition patterns and genotypic variations observed in the study. Addressing the minor points mentioned, defining abbreviations, and offering brief explanations for complex genetic terms, will significantly enhance the overall clarity and accessibility of the results.

4. Discussion

Examples to illustrate the suggested improvements for the discussion section:

1. Functional Significance of Genomic Mutations

   -Original:”The study identified mutations in key genes related to efflux pumps and glycosylation pathways”.

   - Improved: The study identified mutations in efflux pump genes such as MexAB-OprM regulators. These mutations are known to enhance the bacteria's resistance mechanisms by limiting antibiotic efflux, a crucial aspect of resistance development.

2. Addressing Limitations with Bias Consideration:

   - Original: “The genotypic characterization was not performed, which is a limitation.”

   - Improved: While genotypic characterization was not conducted due to resource constraints, it's important to acknowledge that this omission might introduce biases in our understanding of the genetic basis of resistance development. Future studies should consider addressing this limitation to provide a more comprehensive analysis.

3. Practical Implications of Proposed Treatments:

   - Original: “Short-duration treatments with imipenem/relebactam should be evaluated”.

   - Improved: While proposing short-duration treatments with imipenem/relebactam, it's crucial to consider the practical challenges of administering such regimens in clinical settings. Factors such as patient compliance, dosing schedules, and potential side effects should be thoroughly evaluated to determine the feasibility and effectiveness of this approach.

4. Citing Implications of Genomic Mutations:

   - Original:”Genomic mutations were found in genes related to efflux pumps.”

   - Improved:Genomic mutations were detected in genes associated with efflux pumps, such as MexXY/OprM. Previous studies have shown that mutations in these genes can lead to increased efflux activity, contributing to antibiotic resistance.

By incorporating these improvements, the discussion section can provide a more detailed, nuanced, and credible analysis of the research findings.

5. Materials and Methods

The Materials and Methods section of the research paper is well-structured and provides a clear overview of the experimental procedures and techniques used in the study. The methods are described in sufficient detail, allowing readers to understand and potentially replicate the experiments. The section includes information about bacterial strains, experimental procedures, genotypic characterization, bioinformatic analyses, antibiotic susceptibility testing, and statistical analysis.

1. Bacterial Strains (4.1)

   - The source of the bacterial strains is clearly mentioned.

   - It is important to specify any relevant characteristics of the clinical isolates used, such as patient demographics or infection types, if available.

2. In vitro Generation of Resistant Mutants (4.2)

   - The procedure for generating resistant mutants is well-explained, including details about inoculum size, antibiotic concentrations, and incubation conditions.

   - It would be helpful to mention how the stability of the mutations was ensured and confirmed in the generated mutants over the course of the experiment.

3. Antibiotic Susceptibility (4.3)

   - The methods for antibiotic susceptibility testing are described clearly, including the specific antibiotics tested and the protocol followed.

   - Make sure to mention the units used for MIC values (e.g., mg/L) to provide precise information to readers.

4. Statistical Analysis (4.5):

   - The statistical methods used for data analysis are appropriate for the type of data presented.

   - Specify whether the statistical tests used were one-tailed or two-tailed and justify the choice based on the study design.

Author Response

  1. Abstract

The abstract presents a focused study on antibiotic resistance in Pseudomonas aeruginosa, a significant clinical concern. The research aimed to develop an in vitro model using clinical isolates of P. aeruginosa to compare the ability of imipenem and imipenem/relebactam in generating resistant mutants, along with a genotypic analysis to understand the genetic changes under selective pressure. Results indicate that isolates acquired resistance to Imipenem within an average of 6 days, whereas resistance to imipenem/relebactam occurred in 12 days (p-value=0.004). After 30 days, 75% of isolates had a minimum inhibitory concentration (MIC) > 64 mg/L for imipenem, and 37.5% for imipenem/relebactam (p-value=0.077). Notably, the addition of relebactam delayed resistance to imipenem and limited cross-resistance to other beta-lactams. Specific genetic alterations, particularly in glycosylation pathways, transcriptional regulation, histidine kinase response, porins, and efflux pumps, were identified. The study highlights the potential clinical significance of these findings, suggesting that stewardship programs should evaluate the implications for controlling multi-drug resistance in P. aeruginosa. Consideration of study limitations, such as the in vitro nature of the experiment and its applicability to clinical contexts, would further strengthen the abstract.

Response:

The abstract has been improved as suggested by the reviewer:

“The clinical relevance of this phenomenon, which has the limitation that it has been performed in vitro, should be evaluated by stewardship programs in clinical practice, as it could be useful in controlling multi-drug resistance in P. aeruginosa.”

  1. Introduction

The introduction provides a concise overview of the antibiotic resistance issue, focusing on the critical case of Pseudomonas aeruginosa. It effectively emphasizes the urgent need for innovative therapeutic strategies due to the pathogen's adaptability and antibiotic resistance. The link between extended antibiotic usage and the emergence of resistant mutants is clearly articulated, highlighting the study's significance. To enhance clarity, briefly mentioning successful examples of stewardship initiatives and explicitly outlining the unique contribution of this research would add depth. Creating a smoother transition between topics could further improve the introduction's flow, making it more accessible and engaging for a wider readership.

Eg.

  • Original

“The persistent and repeated use of antimicrobials in these critical units exerts selective pressure, hastening the emergence of resistant mutants [3]. Moreover, the transfer of resistance genes among these mutants can lead to the creation of "superbugs" – bacterial strains resistant to most clinically used antimicrobials [4]. In response to this escalating crisis, stewardship initiatives are implementing diverse measures, ranging from rapid microbiological diagnostics to tailored antibiotic therapies, aiming to curb the rise of antibiotic resistance [5,6].”

  • Revised with a Smoother Transition

“The persistent and repeated use of antimicrobials in these critical units exerts selective pressure, hastening the emergence of resistant mutants [3]. Moreover, the transfer of resistance genes among these mutants can lead to the creation of "superbugs" – bacterial strains resistant to most clinically used antimicrobials [4]. Recognizing the urgency of the situation, healthcare providers and researchers have implemented stewardship initiatives. These initiatives encompass a wide range of measures, from the swift diagnosis through rapid microbiological techniques to tailored antibiotic therapies. By understanding these challenges and the ongoing efforts to address them, our study delves into the intricate dynamics of antibiotic resistance in P. aeruginosa, shedding light on crucial areas that demand exploration.”

 As it connects the discussion of superbugs and stewardship initiatives, leading seamlessly into the purpose and relevance of the current study. This approach helps readers follow the logical progression of the introduction more easily.

Response:

The paragraph has been modified as stated by the reviewer:

The persistent and repeated use of antimicrobials in these critical units exerts selective pressure, hastening the emergence of resistant mutants [3]. Moreover, the transfer of resistance genes among these mutants can lead to the creation of "superbugs" - bacterial strains resistant to most clinically used antimicrobials [4]. Recognizing the urgency of the situation, healthcare providers and researchers have implemented stewardship initiatives. These initiatives encompass a wide range of measures, from the swift diagnosis through rapid microbiological techniques to tailored antibiotic therapies. By understanding these challenges and the ongoing efforts to address them, our study delves into the intricate dynamics of antibiotic resistance in P. aeruginosa, shedding light on crucial areas that demand exploration ]. In response to this escalating crisis, stewardship initiatives are implementing diverse measures, ranging from rapid microbiological diagnostics to tailored antibiotic therapies, aiming to curb the rise of antibiotic resistance [5,6].

3.Results

The results section of the research paper is well-structured, providing a clear and detailed account of the experiments conducted. The data are presented logically, and the use of tables and figures enhances the overall clarity of the results. The authors have appropriately utilized statistical analyses, supporting the presented data with relevant p-values to indicate the significance of the findings. However, it is important to explicitly mention the methods employed for the statistical analysis to ensure transparency and reproducibility.

  • Genotypic Characterization and Biological Significance

The genotypic characterization section is detailed and informative, shedding light on the genetic basis of resistance. The description of the pathways affected by mutations is clear. However, considering the complexity of genetic terminology, providing brief explanations or references for readers less familiar with these terms could enhance the accessibility of the content. Moreover, it would be valuable to include a brief concluding paragraph summarizing the key findings of the genotypic characterization, helping readers grasp the implications of the genetic mutations in the context of antibiotic resistance.

In summary, the results section provides a detailed account of the resistance acquisition patterns and genotypic variations observed in the study. Addressing the minor points mentioned, defining abbreviations, and offering brief explanations for complex genetic terms, will significantly enhance the overall clarity and accessibility of the results.

Response:

Abbreviations, explanations of genomic terms and a short summary paragraph have been included.

“ After manual curation we found two OXA-50 (oxacilinase, β.lactamase) variants (OXA-846 and OXA-488) and PDC (Purified β-lactamase) variants (PDC-127 and PDC-34) in both strains, these genes are intrinsic resistance genes in P. aeruginosa (Table S3).”

“Imipenem resistant mutants presented 2 and 9 single nucleotide polymorphisms (SNPs, changes of one nucleotide), compared to their respective susceptible strains (Table 2). We evaluated the effect of the detected SNPs on coding sequences, that is, missense (the SNP’s produce that a different amino acid being incorporated into the structure of the protein), frameshift (reading frame changes, resulting in a protein that is completely different from the original), and stop codon mutations resulting in a smaller, usually non-functional protein, as well as in non-coding regions, which may alter the transcription of the closest gene.  While strain 8247 showed only significant variations on the gene ydphP_1, in the other strain (2718), seven were the genes in which non-synonymous variations were pro-duced (phoQ, gyrB, wecA, sasA_7, nuoM, oprD_7, pilA). On the other hand, imipenem/relebactam resistance mutant showed 5 and 18 total SNP’s when compared with the sensible strain. In this case, genes with significant variants were rne, sasA_14, qseC_2 in 8247 strain, and tyrR_1, rstB, barA_1, zrarR_4, pglF, rne and rnr, in 2718 strain.  Strain 8247 showed the convergent variation in gene tagO when treated with both treat-ments, so this SNP’s could be shared in the imipenem and imipenem/relebactam resistance acquisition mechanism.”

“In summary, the findings indicate that the selective pressure exerted by prolonged exposure to imipenem involves changes at the genotypic level in several biological pathways, and when relebactam is added, the changes also affect the transcriptional mechanism.

  1. Discussion

Examples to illustrate the suggested improvements for the discussion section:

  1. Functional Significance of Genomic Mutations

   -Original:”The study identified mutations in key genes related to efflux pumps and glycosylation pathways”.

   - Improved: The study identified mutations in efflux pump genes such as MexAB-OprM regulators. These mutations are known to enhance the bacteria's resistance mechanisms by limiting antibiotic efflux, a crucial aspect of resistance development.

  1. Citing Implications of Genomic Mutations:

   - Original:”Genomic mutations were found in genes related to efflux pumps.”

   - Improved:Genomic mutations were detected in genes associated with efflux pumps, such as MexXY/OprM. Previous studies have shown that mutations in these genes can lead to increased efflux activity, contributing to antibiotic resistance.

These two corrections has been including together. This paragraph has been modified

In line, the study identified mutations in efflux pump genes such as MexAB-OprM regu-lators. These mutations are known to enhance the bacteria's resistance mechanisms by limiting antibiotic efflux, a crucial aspect of resistance development.this study identified mutations in OprD, an imipenem specific porin, and also in several regulators of efflux pumps.

  1. Addressing Limitations with Bias Consideration:

   - Original: “The genotypic characterization was not performed, which is a limitation.”

   - Improved: While genotypic characterization was not conducted due to resource constraints, it's important to acknowledge that this omission might introduce biases in our understanding of the genetic basis of resistance development. Future studies should consider addressing this limitation to provide a more comprehensive analysis.

Response:

It has been modified

Genotypic characterization was not conducted on all strains due to resource con-strains, so it's important to acknowledge that this omission might introduce biases in our understanding of the genetic basis of resistance development. Future studies should con-sider addressing this limitation, by increasing the number of strains sequenced, to provide a more comprehensive analysis. The genotypic characterisation of this work was not per-formed on the 8 strains studied, as. This genomic”

  1. Practical Implications of Proposed Treatments:

   - Original: “Short-duration treatments with imipenem/relebactam should be evaluated”.

   - Improved: While proposing short-duration treatments with imipenem/relebactam, it's crucial to consider the practical challenges of administering such regimens in clinical settings. Factors such as patient compliance, dosing schedules, and potential side effects should be thoroughly evaluated to determine the feasibility and effectiveness of this approach.

Response:

It has been modified

“Based on the data obtained, the short duration treatments with the imipenem/relebactam, even if the microorganism is susceptible to carbapenems, could to prevent the appearance of resistance during treatment. However, while proposing short-duration treatments with imipenem/relebactam, it's crucial to consider the practical challenges of administering such regimens in clinical settings. Factors such as patient compliance, dosing schedules, and potential side effects should be thoroughly evaluated to determine the feasibility and effectiveness of this approach.”

By incorporating these improvements, the discussion section can provide a more detailed, nuanced, and credible analysis of the research findings.

All the changes proposed by the evaluator have been carried out.

  1. Materials and Methods

The Materials and Methods section of the research paper is well-structured and provides a clear overview of the experimental procedures and techniques used in the study. The methods are described in sufficient detail, allowing readers to understand and potentially replicate the experiments. The section includes information about bacterial strains, experimental procedures, genotypic characterization, bioinformatic analyses, antibiotic susceptibility testing, and statistical analysis.

  1. Bacterial Strains (4.1)

   - The source of the bacterial strains is clearly mentioned.

   - It is important to specify any relevant characteristics of the clinical isolates used, such as patient demographics or infection types, if available.

Response:

The initial samples from which the strains were isolated has been added.

“All the strains were susceptible to imipenem and imipenem/relebactam (EUCAST 2023 criteria), isolated from respiratory, urine or wound exudate samples during 2020.”

  1. In vitro Generation of Resistant Mutants (4.2)

   - The procedure for generating resistant mutants is well-explained, including details about inoculum size, antibiotic concentrations, and incubation conditions.

   - It would be helpful to mention how the stability of the mutations was ensured and confirmed in the generated mutants over the course of the experiment.

Response:

It has been added in the text:

“…eight mutants resistant to imipenem, and eight mutants resistant to imipenem/relebactam were obtained, to ensure the stability of the mutations generated, the strains were re-seeded five consecutive times on antibiotic-free medium.”

  1. Antibiotic Susceptibility (4.3)

   - The methods for antibiotic susceptibility testing are described clearly, including the specific antibiotics tested and the protocol followed.

   - Make sure to mention the units used for MIC values (e.g., mg/L) to provide precise information to readers.

Response:

The manuscript has been revised, ensuring that the units of antibiotic concentration are correct.

  1. Statistical Analysis (4.5):

   - The statistical methods used for data analysis are appropriate for the type of data presented.

   - Specify whether the statistical tests used were one-tailed or two-tailed and justify the choice based on the study design.

Response:

The explanation and justification have been included in the text

“The tests used are two tailed (split the significant level and applied in both direction), suitable for determining whether there is any difference between the groups and a p-value<0.05”

Reviewer 4 Report

Comments and Suggestions for Authors

Dear Authors,

 please find my review of your manuscript.

Review of the article: Role of Relebactam in the Antibiotic Resistance Acquisition in P.              

aeruginosa: in vitro study

Authors: Maria Paz Ventero , Jose M Haro-Moreno , Carmen Molina-Pardines * , Antonia Sanchez-Bautista , Celia García-Rivera , Vicente Boix , Esperanza Merino , Mario López-Pérez , Juan Carlos Rodriguez

Manuscript ID: antibiotics-2658806

Submitted to the Journal: Antibiotics (ISSN 2079-6382); Section: Mechanism and Evolution of Antibiotic Resistance; Special Issue: Antibiotics Resistance and Molecular Epidemiology of Carbapenem-Resistance Bacteria

The article, titled " Role of Relebactam in the antibiotic resistance acquisition in P.  

aeruginosa: in vitro study," investigates the impact of relebactam on the acquisition of resistance by Pseudomonas aeruginosa. 

Understanding the role of relebactam and other adjuvants in antibiotic resistance acquisition is essential for improving the effectiveness of antibiotics, reducing the development of resistance, and ultimately enhancing patient care and public health outcomes. The study aimed to generate significant new data on antibiotic resistance development. However, the manuscript in its current form requires major and minor editorial as well as editorial changes. Below I present my major and minor concerns regarding the manuscript.

1. The full name Pseudomonas aeruginosa should be listed in the title. 

2. Line 21 and throughout the rest of the manuscript: pvalue should be written as "P value."

3. The acronym has been coined by Rice LB. J Infect Dis. 2008 Apr 15;197(8):1079-81. doi: 10.1086/533452. Not by WHO.

4. Line 44: consider changing “clinical routine” to “clinical practice”

5. Line 55: explain PA01

6. Line: in vitro write in italics

7. The section “Introduction” should present more information about the relebactam, NGS, and short and long reads. Not every reader is familiar with NGS techniques.

8. Section Materials and Methods should be before section Result, if the publisher requests applied order, underline it at the beginning of the section Results.

9. Line 140: explain what PDC means.

10. Lines 145-155: present results of sequencing, SNPs, its position, and effect on coding sequence.

11. Lines 167-169/Table 2: second column is without name

12. Lines 167-169/Table 2: unify style of writing

13. Lines 167-169/Table 2: genes names should be written in italics

14. Lines 167-169/Table 2: is there any reason for bolding some of the lines? If yes, please explain it in the manuscript. To some extent, it is presented in the legend below the table, but in its current form, it is difficult to understand.

15. Throughout the manuscript a space between “word” and [reference] must be applied, eg. lines 192, 205, 207.

16. The section 4. Materials and Methods:

a. Explain the ARIBA acronym

b. Present the drug susceptibility of the strains used for study and information about them, eg. the source of isolation, year of isolation, etc.

c. Provide more information regarding sample preparation for sequencing and MLST typing.

d. The data presented in the supplementary materials should be included in the manuscript and described.

17. Line 194: P. fluorescens – italicize

18. Discussion should be elaborated.

Summary 

In its current form, the manuscript cannot be considered suitable for publication. Significant revisions are necessary to address the issues mentioned above and enhance the overall quality 

and clarity of the paper. After substantial editing and improvement, the manuscript may be reconsidered for publication after an additional review process.

Sincerely yours

Reviewer

Author Response

Dear Authors,

Pease find my review of your manuscript.

Review of the article: Role of Relebactam in the Antibiotic Resistance Acquisition in P. aeruginosa: in vitro study

Authors: Maria Paz Ventero , Jose M Haro-Moreno , Carmen Molina-Pardines * , Antonia Sanchez-Bautista , Celia García-Rivera , Vicente Boix , Esperanza Merino , Mario López-Pérez , Juan Carlos Rodriguez

Manuscript ID: antibiotics-2658806

Submitted to the Journal: Antibiotics (ISSN 2079-6382); Section: Mechanism and Evolution of Antibiotic Resistance; Special Issue: Antibiotics Resistance and Molecular Epidemiology of Carbapenem-Resistance Bacteria

The article, titled " Role of Relebactam in the antibiotic resistance acquisition in P. aeruginosa: in vitro study," investigates the impact of relebactam on the acquisition of resistance by Pseudomonas aeruginosa.

Understanding the role of relebactam and other adjuvants in antibiotic resistance acquisition is essential for improving the effectiveness of antibiotics, reducing the development of resistance, and ultimately enhancing patient care and public health outcomes. The study aimed to generate significant new data on antibiotic resistance development. However, the manuscript in its current form requires major and minor editorial as well as editorial changes. Below I present my major and minor concerns regarding the manuscript.

  1. The full name Pseudomonas aeruginosa should be listed in the title.

Response:

The title has been modified according the reviewer suggests

  1. Line 21 and throughout the rest of the manuscript: pvalue should be written as "P value."

Response:

The manuscript has been revised, and all pvalue have been change by P value

  1. The acronym has been coined by Rice LB. J Infect Dis. 2008 Apr 15;197(8):1079-81. doi: 10.1086/533452. Not by WHO.

Response:

The right reference has been added.

“the acronym "ESKAPE" by Rice L.B in 2008 [1] the WHO,”

  1. Line 44: consider changing “clinical routine” to “clinical practice”

Response:

This paragraph has been amended in accordance with a suggestion from another reviewer, the final paragraph is:

In addition, this pathogen often develops antibiotic susceptibility changes during treatment, especially in intensive care units and immunosuppressed patients [2,3].  The persistent and repeated use of antimicrobials in these critical units exerts selective pressure, hastening the emergence of resistant mutants [4]

  1. Line 55: explain PA01

Response:

This has been explained

“of P. aeruginosa PAO1 (the most commonly used strain for research)”

  1. Line: in vitro write in italics

Response:

It has been fully revised and corrected.

  1. The section “Introduction” should present more information about the relebactam, NGS, and short and long reads. Not every reader is familiar with NGS techniques.

Response:

Information on relebactam and both NGS techniques was included throughout the introduction.

  1. Section Materials and Methods should be before section Result, if the publisher requests applied order, underline it at the beginning of the section Results.

Response:

The publisher's template has the Materials and Methods section at the end of the manuscript. As requested by the reviewer, we have indicated it at the beginning of the Results.

  1. Line 140: explain what PDC means.

Response: 

It have been explained:

“After manual curation we found two OXA-50 (oxacilinase, β.lactamase) variants (OXA-846 and OXA-488) and PDC (Purified β-lactamases)

  1. Lines 145-155: present results of sequencing, SNPs, its position, and effect on coding sequence.

Reponse:

The authors think that this might be too much information to include in the text, but agree with the reviewer that this information is important for the reader, so this data has been included as a supplementary table (S2).

  1. Lines 167-169/Table 2: second column is without name

Response:

The column name has been included

  1. Lines 167-169/Table 2: unify style of writing
  2. Lines 167-169/Table 2: is there any reason for bolding some of the lines? If yes, please explain it in the manuscript. To some extent, it is presented in the legend below the table, but in its current form, it is difficult to understand.

The explanation in the footnote has been improved

"Underlined sentences: Pathways affected in both type of mutants

Bold sentences: Pathways affected only in imipenem resistant mutants

No special format sentences: Pathways affected only in imipenem/relebactam resistant mutants"

  1. Lines 167-169/Table 2: genes names should be written in italics

Response:

It has been fully revised and corrected.

  1. Throughout the manuscript a space between “word” and [reference] must be applied, eg. lines 192, 205, 207.

Response:

It has been fully revised and corrected.

  1. The section 4. Materials and Methods:
  2. Explain the ARIBA acronym

Response:

It has been explained in the text

“…using ARIBA (Antimicrobial Resistance Identification By Assembly) [45]”

  1. Present the drug susceptibility of the strains used for study and information about them, eg. the source of isolation, year of isolation, etc.

Response:

The drug susceptibility of the initial strains are indicated in S1. The source of isolation and the year have been included in the manuscript.

“All the strains were susceptible to imipenem and imipenem/relebactam (EUCAST 2023 criteria), isolated from respiratory, urine or wound exudate samples during 2020.”

  1. Provide more information regarding sample preparation for sequencing and MLST typing.

Response:

The kit used for the library preparation has been added. 

"Strains were sequenced by Illumina NextSeq 500/550 High Output Kit v2.5 -- 300 Cycles, using the kit DNA Prep distributed by Illumina"

  1. The data presented in the supplementary materials should be included in the manuscript and described.

Response:

The table S2 and S3 have been included in the text as table 2 and table 3. However, the authors think that the other tables would be better presented as Supplementary materials due to their size and type of information, whether the editor ask for it, the authors could modified it.

  1. Line 194: P. fluorescens – italicize

Response:

The format of P. fluorescens have been change, now is in italic. 

  1. Discussion should be elaborated.

Response:

Some paragraphs of the discussion have been modified at his and the other reviewer's suggestion.

Summary

In its current form, the manuscript cannot be considered suitable for publication. Significant revisions are necessary to address the issues mentioned above and enhance the overall quality and clarity of the paper. After substantial editing and improvement, the manuscript may be reconsidered for publication after an additional review process.

The authors have considered and applied the required revisions. 

Round 2

Reviewer 1 Report

Comments and Suggestions for Authors

While the authors' modifications based on my previous comments have notably improved the manuscript, I remain concerned about their response to comment 1. Although steps were taken to ensure mutation stability and gDNA isolation from single colonies, relying solely on a single colony for gDNA isolation may limit the representation of mutations, potentially affecting the diversity observed among different colonies. I strongly emphasize the critical need for biological replicates in these experiments to ensure a more comprehensive assessment of mutational variations and to strengthen the reliability and validity of the study's conclusions.

Author Response

While the authors' modifications based on my previous comments have notably improved the manuscript, I remain concerned about their response to comment 1. Although steps were taken to ensure mutation stability and gDNA isolation from single colonies, relying solely on a single colony for gDNA isolation may limit the representation of mutations, potentially affecting the diversity observed among different colonies. I strongly emphasize the critical need for biological replicates in these experiments to ensure a more comprehensive assessment of mutational variations and to strengthen the reliability and validity of the study's conclusions.

Response

The authors agree with the reviewer, this is a limitation of the study that we couldn’t assess because the analysis of different pure colonies from each strain increase the complexity of the study and also the costs, and the funding are limited. Taking into account the reviewer's comment, the authors have included this point in the limitations paragraph included in the discussion. Furthermore, the data obtained in this work are the starting point for future studies to analyse this complex phenomenon.

“Genotypic characterization was not conducted on all strains, and the gDNA was isolated from one single colony for each strain, due to resource constrains, so it's important to acknowledge that this omission might introduce biases in our understanding of the genetic basis of resistance development. Future studies should consider addressing this limitation, by increasing the number of strains sequenced and isolation the gDNA’s from different colonies from the each strain, to provide a more comprehensive analysis.”

Reviewer 3 Report

Comments and Suggestions for Authors

None

Author Response

Thank you for your revision.

Reviewer 4 Report

Comments and Suggestions for Authors

Dear Authors 

Thank you for the comprehensive revisions you have made to your manuscript, which has significantly enhanced the quality of your work.

However, after a thorough review of your manuscript, I have identified a few remaining minor editorial issues that merit attention. These corrections are as follows:

In Table 2, it appears that the number of isolates is not separated by spaces.

On Line 243, "aminoglycosides (PhoQ)" should have a lowercase "p" in "PhoQ."

Line 336 features the phrase "By other hand." I would suggest revising it to "On the other hand." Alternatively, consider using an alternative phrase such as "Despite that..."

On Line 340, the insertion of the word "has" is recommended between "it" and "been."

Line 357 should be corrected to "these changes cannot be detected."

I believe that these minor adjustments will further refine your manuscript.

Kind regards,

Reviewer

Comments on the Quality of English Language

Line 336 features the phrase "By other hand." I would suggest revising it to "On the other hand." Alternatively, consider using an alternative phrase such as "Despite that..."

On Line 340, the insertion of the word "has" is recommended between "it" and "been."

Line 357 should be corrected to "these changes cannot be detected."

Author Response

In Table 2, it appears that the number of isolates is not separated by spaces.

Response:

The spaces have been added.

On Line 243, "aminoglycosides (PhoQ)" should have a lowercase "p" in "PhoQ."

Response:

This has been corrected.

Line 336 features the phrase "By other hand." I would suggest revising it to "On the other hand." Alternatively, consider using an alternative phrase such as "Despite that..."

Response:

This has been modified.

 “On the other hand, the glycosylation pathway”

On Line 340, the insertion of the word "has" is recommended between "it" and "been."

Response:

This has been modified.

“glycosylation it has been proposed”

Line 357 should be corrected to "these changes cannot be detected."

Response:

This has been modified.

“these changes cannot be detected.”